# ROUTING EXPERTS: LEARNING TO ROUTE DYNAMIC EXPERTS IN EXISTING MULTI-MODAL LARGE LANGUAGE MODELS

**Qiong Wu**[12], **Zhaoxi Ke**[12], **Yiyi Zhou**[12]*, **Xiaoshuai Sun**[12], **Rongrong Ji**[12]

[1] Key Laboratory of Multimedia Trusted Perception and Efficient Computing,
Ministry of Education of China, Xiamen University, 361005, P.R. China.
[2] Institute of Artificial Intelligence, Xiamen University, 361005, P.R. China.
{qiong, kezhaoxi}@stu.xmu.edu.cn, {zhouyiyi, xssun, rrji}@xmu.edu.cn,

## ABSTRACT

Recently, *mixture of experts* (MoE) has become a popular paradigm for achieving the trade-off between modal capacity and efficiency of *multimodal large language models* (MLLMs). Different from previous efforts, we are dedicated to exploring the dynamic experts in existing MLLMs and showing that a standard MLLM can also be a mixture of experts. However, achieving this target is still notoriously challenging. The well-trained MLLMs are more accustomed to the fixed pathway and a drastic change in its inference manner also greatly impedes its performance. To address these issues, we propose a novel dynamic expert routing method for existing MLLMs, termed ***Routing Experts*** (**RoE**), which can achieve example-dependent optimal path routing without obvious structure tweaks. Meanwhile, a new structure sparsity regularization is also introduced to force the well-trained MLLMs to learn more short-cut pathways. In addition, we also address the alignment of the training and inference of MLLMs in terms of network routing. To validate RoE, we apply it to a set of existing MLLMs, including LLaVA-1.5, LLaVA-HR and VILA, and conduct extensive experiments on a bunch of VL benchmarks. The experiment results not only show the effectiveness of our RoE in improving MLLMs' efficiency, but also yield obvious advantages over MoE-LLaVA in both performance and speed, *e.g.*, an average performance gain of 3.3% on 5 benchmarks while being 1.61 times faster. Our code is anonymously released at https://github.com/DoubtedSteam/RoE

## 1 INTRODUCTION

Recently, the great success of *large language models* (LLMs) Radford et al. (2018); Zhang et al. (2022); Bai et al. (2023a); Touvron et al. (2023) attracts an influx of interest in extending them to more modalities, *e.g.*, *vision and language* (VL) Jiang et al. (2020); Luo et al. (2023b); Tong et al. (2024); Zhou et al. (2019). Despite great progress, *multi-modal large language models* (MLLMs) Li et al. (2023b); Dai et al. (2023); Rose et al. (2023); Wang et al. (2024); Koh et al. (2024) also suffer from excessive computation due to the introduction of more modality tokens. For instance, LLaVA Liu et al. (2023b) requires 6.15 times more computation than its unimodal inference on ScienceQA Lu et al. (2022). Inspired by the progress of LLMs Radford et al. (2018); Zhang et al. (2022); Touvron et al. (2023), recent efforts Bai et al. (2023a); Ainslie et al. (2023); Jiang et al. (2024); Raposo et al. (2024) are also devoted to exploring new MLLMs with a *Mixture-of-Experts* (MoE) structure, thereby archiving a good trade-off between model capacity and inference efficiency.

Different from these pioneers Gou et al. (2023); Shen et al. (2023); Lin et al. (2024), we focus on exploring the dynamic experts in MLLMs that already exist and show that a well-trained common MLLM can also be a mixture of experts. The motivation is akin to MoE, that is, LLMs or MLLMs need enough parameter capacity to meet *scaling law* Kaplan et al. (2020), but it is evident that the entire model is often redundant for specific tasks, especially the easy ones Wu et al. (2024). For

---

*Corresponding Author.

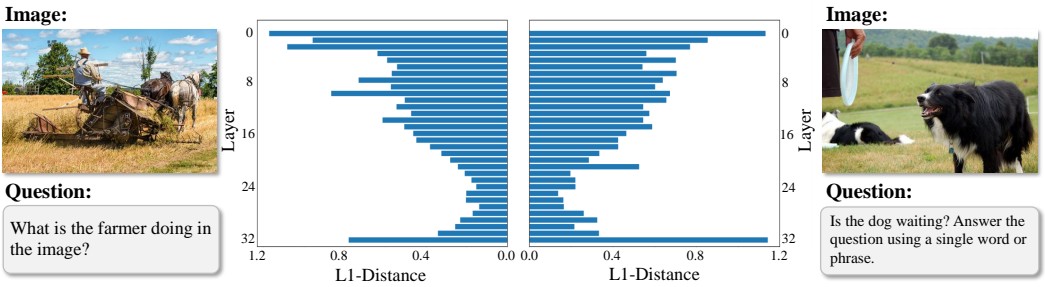

Figure 1: The visualization of the $l1$-distances between the input and output features of each layer of LLaVA-7B Liu et al. (2023b). A lower $l1$-distance[1] indicates that this layer has less impact on the feature update of this example, which also suggests that it is not that important during inference. For two examples, the contributions of different layers are also different.

instance, the advanced MLLMs like LLaVA-1.5 Liu et al. (2023a) exhibit much stronger generalization capability than previous vision-language (VL) models Li et al. (2019); Kim et al. (2021); Dou et al. (2022); Gao et al. (2023), but is still on par with the bespoke ones Lu et al. (2019); Kim et al. (2021) with much smaller parameter sizes on the benchmarks like VQAv2 Goyal et al. (2017).

However, in terms of methodology, we are keen to explore the dynamic and sparse structures of MLLMs that already exist, rather than building a new sparse model like previous MoE methods Shen et al. (2023); Lin et al. (2024). We observe that the activations of MLLMs' different layers for the examples are distinct. As shown in Fig. 1, some layers barely contribute to the transformation and reasoning of a given example. This finding implies that the inherent knowledge of common MLLMs is likely to be distributed as in MoE models Eigen et al. (2013), indicating the feasibility of routing the expert subnetworks in an already existing MLLM.

However, achieving this target is still intractable. In particular, we aim to adaptively skip the less important layers of MLLMs for each example, thereby achieving better efficiency, as shown in Fig. 2. Although intuitive, this attempt at MLLMs still suffers from several challenges. The first one is the feature gap that occurs in dynamic inference. Unlike previous dynamic models Lin et al. (2024); Jiang et al. (2024); Sun et al. (2024); Shen et al. (2024); Luo et al. (2024b), which are mostly trained from scratch and well accommodate dynamic inference, this layer-wise skipping will make MLLMs encounter a drastic change in feature space during inference, greatly limiting its performance upper-bound. Meanwhile, how to make MLLMs choose a short-cut pathway is also difficult. Since MLLMs are already end-to-end well trained, they usually prefer not to skip under the default training objectives. In addition, existing MLLMs Dai et al. (2023); Liu et al. (2023a); Lin et al. (2023); Liu et al. (2023b); Zhou et al. (2024) often organize multiple examples as a multi-turn conversation for efficient training, which however contradicts the dynamic routing of each example. Overall, these ingredients greatly hinder the achievement of dynamic routing in existing MLLMs.

To address these issues, we propose an innovative dynamic routing method for MLLMs, termed *Routing Experts* (RoE). RoE regards each layer of MLLMs as an independent expert, and its objective is to find out and connect the important ones as an optimal routing path for each example. In practice, RoE uses path routers to decide whether to skip layers. To compensate the issue of the feature gap, we introduce the adapter-based connections to replace the less important layers, which are easy to deploy and can well serve feature transformations Wu et al. (2024). To optimize RoE, we also propose a novel sparsity regularization to encourage the learning of sparse and diverse pathway routing. Combined with this objective, the simple yet effective routing tokens are further proposed to facilitate the optimization of RoE in multi-turn conversations, addressing the issue of misalignment between training and inference. With these innovative designs, RoE shows that a standard and well-trained MLLM can also be a mixture of experts.

To validate RoE, we apply it to a set of advanced MLLMs, including LLaVA-1.5 Liu et al. (2023a), LLaVA-HR Luo et al. (2024c) and VILA Lin et al. (2023), on 10 competitive VL benchmarks, including VQA2.0 Goyal et al. (2017), GQA Hudson & Manning (2019), TextVQA Singh et al. (2019), POPE Li et al. (2023c), MME Fu et al. (2023), and MM-Vet Yu et al. (2023). The experimental results show that our RoE can greatly speed up the inference of common MLLMs, while still maintaining their competitive performance on various benchmarks. For instance, our RoE improves the inference speed of LLaVA-1.5 by 21.3% without performance drops on most benchmarks. Com-

pared with the previous MoE-based method, *e.g.*, MoE-LLaVA Lin et al. (2024), RoE not only has better performance on all benchmarks, but also exhibits faster inference speed, *e.g.*, 6.77 v.s. 4.95 examples per second for traditional VL benchmarks on average.

Overall, our contributions are three-fold:

- We present a novel attempt of dynamic routing in existing MLLMs, termed ***Routing Experts*** (RoE), which aims to transform existing MLLMs into a mixture of experts without obvious structure tweaks.
- To achieve effective and efficient RoE for the well-trained MLLMs, we introduce adapter-based skip connection to alleviate the feature gap problem and a novel sparsity regularization to help MLLMs learn dynamic and sparse inference. Besides, the routing token design also aligns the training and inference of RoE-MLLMs.
- On ten highly-competitive benchmarks, RoE can significantly improve the efficiency of three advanced MLLMs while retaining similar or even better performance.

## 2 RELATED WORK

### 2.1 MIXTURE-OF-EXPERTS

Mixture-of-Experts (MoE) Eigen et al. (2013); Jain et al. (2024); Zhao et al. (2024) is a dynamic and sparse paradigm that can achieve a good trade-off between model capability and efficiency. Its main property is that MoE models can dynamically select the most appropriate experts from several candidates for different inputs, thereby improving model efficiency. In terms of methodology, existing MoE models can be categorized into the *soft* and the *hard* ones, respectively. In soft MoE Lepikhin et al. (2020); Fedus et al. (2022); Lin et al. (2024), the model output is a weighted aggregation of the experts with high confidence. For example, MoE-LLaVA Lin et al. (2024) combines outputs from multiple *feedforward networks* (FFNs) to enhance the model capabilities. Mistral-MoE Jiang et al. (2024) uses the outputs of top-two experts for different examples. Although effective, soft MoE is often hard to achieve real speed acceleration as expected, since the inference of all experts needs to be computed. In contrast, hard MoE models Kudugunta et al. (2021); Bao et al. (2022); Zhu et al. (2022); Satar et al. (2022); Wang et al. (2022b); Shen et al. (2023); Li et al. (2023d); Ma et al. (2023); Long et al. (2023) dynamically activate the experts, introducing less additional calculation overhead. For instance, VLMo Bao et al. (2022) and VL-MoE Shen et al. (2023) activate the expert with the highest confidence. PaCE Li et al. (2023d) activates experts according to the predefined token given by the input. Although MoE can select appropriate experts to deal with different inputs, it still uses the same complexity for tasks of different difficulties. In the latest developments, some methods Ainslie et al. (2023); Dotzel et al. (2024); Jaiswal et al. (2024); Raposo et al. (2024); Luo et al. (2024a) introduce experts with different computational overhead to improve the efficiency. For instance, CoLT5 Ainslie et al. (2023) proposes a heavy and light option for each module in a transformer layer. DLO Zhen et al. (2024) expands the Transformer layers vertically and selectively activates some ones from them. However, these MoE models often need to re-design the network structure and train the model from scratch, lacking the effective use of existing MLLMs. Recently, some works introduce dynamic inference. SkipBERT Wang et al. (2022a) improves shallow layers' efficiency in BERT by combining words through n-grams. SmartBERT Hu et al. (2023) combines layer-skipping and early-exit using the [cls] token. MoD Raposo et al. (2024) focuses tokens to take shortcuts according to a certain ratio in each layer. Orthogonal to these works, we focus on exploring the inherent expert structure in MLLMs that already exist.

### 2.2 MULTI-MODAL LARGE LANGUAGE MODEL

Driven by the success of *large language models* (LLMs) Radford et al. (2018); Touvron et al. (2023); Chen et al. (2023c); Bai et al. (2023a); Jiang et al. (2024); Zhang et al. (2024), the research of *multimodal large language models* (MLLMs) Alayrac et al. (2022); Zhu et al. (2023); Bai et al. (2023b); Chen et al. (2023a;b); Li et al. (2023b); Liu et al. (2023b;c); Peng et al. (2023); Luo et al. (2023a; 2024c); Zhou et al. (2024); Zhu et al. (2024) also gains increasing attention recently. The main paradigm of MLLMs is to directly connect the visual encoder and LLM with an additional network. For instance, BLIP-2 Li et al. (2023b) introduces QFormer to bridge the gap between

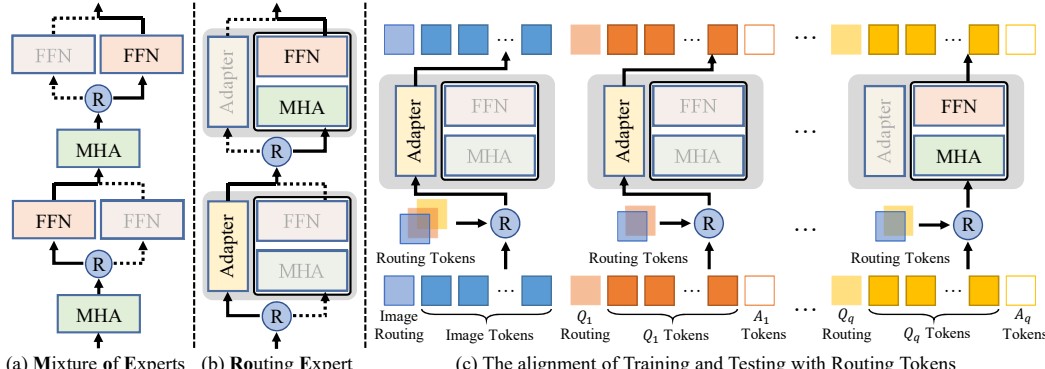

(a) **M**ixture **of E**xperts  (b) **Ro**uting **E**xpert  (c) The alignment of Training and Testing with Routing Tokens

Figure 2: Illustration of the proposed *Routing Experts* (RoE). Existing MoE models (a) often build a new sparse structure with multiple FFNs as experts, and each pathway takes the same computation for all examples. Our RoE (b) aims to explore the expert pathways within the model itself via adapter-based skip connections, realizing dynamic computation for different examples. (c) Routing tokens are used to decide layer-wise path selection, *i.e.*, the adapter-based skip connection or the default Transformer layer. It also serves to align the training and testing of MLLMs.

vision and language modalities, integrating visual tokens into LLMs. Similarly, MINI-GPT4 Zhu et al. (2023) uses a projection layer to map visual features into the semantic space of the LLM. LLaVA Liu et al. (2023b) shares the same paradigm with MINI-GPT, and also proposes a carefully designed training strategy. In terms of network design, MLLMs often use a stack of Transformer decoding layers Bai et al. (2023a); Touvron et al. (2023) for multi-modal inference following LLMs. However, with the introduction of visual tokens, the already high computation of this dense structure is further exacerbated. To address this issue, recent MLLMs like MoE-LLaVA Lin et al. (2024) resort to sparse and dynamic design of MLLMs. However, as mentioned, the computation of multiple paralleled experts still limits the efficiency improvement. Different from these efforts, we aim to explore the dynamic inference of MLLMs to improve efficiency while retaining performance.

## 3 PRELIMINARY

In this section, we first recap the principle of *Mixture of Experts* (MoE) for MLLMs. As shown in Fig. 2(a), existing MoE-MLLMs like *MoE-LLaVA* Lin et al. (2024) often build multiple FFNs as the experts of each layer. During inference, only one or several experts are activated in each layer. In this case, an MoE layer in MLLM $G(\cdot)$ can be defined by

$$\mathbf{x}_{i+1} = \mathbf{x}_i + \sum_{j=1}^{K} G_{ij}(\mathbf{x}_i) \cdot R_{ij}(\mathbf{x}_i), \qquad (1)$$

where $G_{ij}(\cdot)$ denotes the $j$-th expert in the $i$-th layer, and $R_{ij}(\cdot)$ are routing weights predicted by the router $R_i$. $\mathbf{x}_i \in \mathcal{R}^{n \times d}$ represents the inputs for $i$-th layer, where $n$ and $d$ denote the length and dimension. In practice, MoE models only activate a subset of experts according to the input features. Despite effectiveness, each inference still has the same computation for all tasks, which is also redundant for some examples.

Like LLMs Ainslie et al. (2023); Raposo et al. (2024), existing MLLMs are also obviously redundant in many cases. As aforementioned, not all layers equally contribute to the final prediction. In this case, we focus on exploring the dynamic experts in MLLMs that already exist.

Concretely, we regard each layer of an MLLM as an expert, and skip the less important ones to form the routing path $G(x)'$:

$$G' = G_1 \circ G_2 \circ ... \circ G_n, \qquad (2)$$

where $G'(\mathbf{x})$ is the activated subnetwork, and $\{G_1, G_2, ..., G_n\}$ are layers chosen by the router. Its number is smaller than the default length $n$.

However, the absence of some layers in Eq.2 inevitably impedes the feature transformation during inference, especially for the well-trained MLLMs. This issue also makes MLLMs prefer not to skip layers during the training of dynamic routing.

## 4 ROUTING EXPERTS

### 4.1 METHOD

In this paper, we propose an innovative **Routing Experts** (RoE) scheme for the dynamic routing in the existing MLLMs. The objective of RoE is

$$\underset{\boldsymbol{\theta}'}{\arg\min} \, \mathcal{L}\big(G(I, T|[\boldsymbol{\theta}'])\big) + |\boldsymbol{\theta}'|, \tag{3}$$

where $\boldsymbol{\theta}' \subseteq \boldsymbol{\theta}$ is a subset of MLLM, and $|\boldsymbol{\theta}'|$ represents the activated parameters.

As discussed above, a direct skipping is prone to hindering feature transformation, *i.e.*, the feature gap between layers. In this case, we introduce an adapter-based skip connection for MLLMs, and the dynamic expert pathway $G(x)'$ is obtained by

$$G' = M_1 \circ M_2 \circ ... \circ M_n,$$
$$\text{where} \quad M_i = \begin{cases} G_i, R_i(\mathbf{x}_i)_0 > R_i(\mathbf{x}_i)_1, \\ A_i, R_i(\mathbf{x}_i)_0 \leq R_i(\mathbf{x}_i)_1. \end{cases} \tag{4}$$

Here, $M_i$ is the expert of the $i$-th layer, $A_i$ is a lightweight adapter Sung et al. (2022). $R_i$ is a binary routing function to decide whether the $i$-th layer need to be skipped:

$$R(\mathbf{x}_i) = \text{Softmax}(\mathbf{r}_i \mathbf{W}_r), \tag{5}$$

where $\mathbf{r}_i \in \mathcal{R}^{1 \times d}$ is a router token and will be introduced in Sec.4.3.

In terms of the adpater $A_i(\cdot)$, its low-rank network can be defined by

$$\mathbf{x}' = \text{ReLU}(\mathbf{x}\mathbf{W}_d)\mathbf{W}_u, \tag{6}$$

where $\mathbf{W}_d \in \mathbb{R}^{d \times c}$ and $\mathbf{W}_u \in \mathbb{R}^{c \times d}$ are two weight matrices, and $c << d$.

Compared with Eq.2, Eq.4 introduces the use of adapter-based skip connection. Although the adapter involves some computation, but it is much more lightweight than a standard MLLM layer. More importantly, it has been proven to be capable of feature adaption for large models in terms of parameter efficient tuning Sung et al. (2022); Wu et al. (2024).

### 4.2 STRUCTURE SPARSITY REGULARIZATION

Although RoE can cope with the issue of feature gap, the MLLM is still likely to use the entire network to infer examples during and after training, as discussed above.

In this case, we introduce a structure sparsity regularization to facilitate the training of RoE:

$$\mathcal{L}_s = \max(t - \frac{1}{n}\sum_{i=1}^{n} R_i(\mathbf{x}_i)_1, 0), \tag{7}$$

where $n$ denotes the number of layers, and $t$ is a predefined rate of skipped layers. This regularization term can force the model to meet the target skip rate during training.

However, Eq.7 does not consider the varying difficulties of examples. Intuitively, the more difficult the example the more complex inference is required. Thus, we weight the regularization based on the tuning loss of each example. RoE uses the prediction loss value as an indicator of difficulty, and combines it with the overall optimization objective, defined by

$$\mathcal{L} = \mathcal{L}_t + \alpha e^{-|\mathcal{L}_t|}\mathcal{L}_s, \tag{8}$$

where $\mathcal{L}_t$ is the default objective, and $|\mathcal{L}_t^{(k)}|$ demotes the loss value for the example. And $\alpha$ is a hyper-parameter to balance the performance and efficiency. With Eq. 8, the routing for the simpler examples will be optimized more by the sparsity regularization.

Although Eq. 8 is well designed, its actual effectiveness is still greatly limited by the training scheme of existing MLLMs. To explain, existing MLLMs Liu et al. (2023a); Lin et al. (2023); Luo et al. (2024c) often combine multiple VL examples in one multi-turn conversation as a single example, thereby speeding up training. Thus, this type of training examples will share a common routing strategy for parallel computation under the default setting. However, each conversation in the training example should be encouraged to learn an unique routing path to maximize the benefits of our sparsity regularization. To address this issue, we further introduce the design of routing token.

### 4.3 ROUTING TOKEN AND THE ALIGNMENT OF TRAINING AND TESTING

As discussed above, recent MLLMs like LLaVA Liu et al. (2023a) often combine multiple VL examples as one multi-turn conversation. During training, the answers of all examples are predicted and optimized in parallel, which poses a practical issue for the training of dynamic routing, *i.e.*, *how to train and optimize RoE for all examples at the same time?*

To overcome this problem, we introduce the design of routing token for RoE-MLLMs. In practice, we insert the routing tokens $\mathbf{r}_i$ into the input sequence of each example:

$$\mathbf{x}_i = [\mathbf{r}_i^{(0)}, \mathbf{I}_i, \mathbf{r}_i^{(1)}, \mathbf{Q}_i^{(1)}, \mathbf{A}_i^{(1)}, \mathbf{r}_i^{(2)}, \mathbf{Q}_i^{(2)}, \mathbf{A}_i^{(2)}, ..., \mathbf{r}_i^{(q)}, \mathbf{Q}_i^{(q)}, \mathbf{A}_i^{(q)}], \tag{9}$$

where $\mathbf{I} \in \mathbb{R}^{n_v \times d}$ represent the visual tokens. $(\mathbf{Q}_i^{(k)}, \mathbf{A}_i^{(k)})$ is the question-answer pair, where $\mathbf{Q}_i^{(k)} \in \mathbb{R}^{n_q^{(k)} \times d}$ and $\mathbf{A}_i^{(k)} \in \mathbb{R}^{n_a^{(k)} \times d}$ are the question and answer tokens. The inserted routing tokens are learnable vectors that can aggregate information from the corresponding question $\mathbf{Q}_i^k$. And $\mathbf{r}_i^{(0)}$ is the router token for the image. Then, the routing for $j$-th question-answer pair is predicted by

$$R_i(\mathbf{x}_i)^{(j)} = \text{Softmax}(\frac{1}{\tau}[\mathbf{r}_i^{(0)}, \mathbf{r}_i^{(j)}]\mathbf{W}_r), j > 0, \tag{10}$$

where $[\cdot, \cdot]$ denotes concatenation, $\mathbf{W}_r \in \mathbb{R}^{2d \times 2}$ is a weight matrix, and $\tau$ is the temperature that decreases as training progresses. For the image sequence, its routing weights are computed by

$$R_i(\mathbf{x}_i)^{(0)} = \text{Softmax}(\frac{1}{q\tau} \sum_{k=1}^{q} [\mathbf{r}_i^{(0)}, \mathbf{r}_i^{(k)}]\mathbf{W}_r), \tag{11}$$

where $q$ is the number of questions.

This design allows each question to engage in the prediction of its specific expert pathway while maintaining training efficiency. In this case, we can align the training and inference of MLLMs for dynamic routing.

### 4.4 THE TRAINING SCHEME OF RoE

In this paper, we also carefully design the training scheme of RoE consisting of three main stages.

**Stage 1: Adapter warmup.** We first optimize the adapter-based skip connections, thereby making them be capable of feature transformation. In particular, we will randomly select the default layers and some adapter connections as the expert network according to a predefined sparsity target. To reduce the difficulty of optimization, we will freeze the entire MLLMs and only update the adapters.

**Stage 2: Router warmup.** When the adapter connections are well trained, we begin to optimize the routers for path routing. At this stage, the MLLM is still frozen, and both adapters and routers are trained. Meanwhile, the structure sparsity regularization of RoE begins to be used during training.

**Stage 3: Instruction tuning.** Lastly, we updated the entire RoE and MLLM for the instruction tuning, and the training objectives include the sparsity regularization and the default ones.

Although RoE involves three training stages, the actual expenditure is cost-effective. Above all, RoE does not require the expensive VL alignment pertaining. Meanwhile, the learnable parameter spaces of the adapters and routers are small, so they can be quickly optimized by a few SFT steps. Similarly, the final instruction tuning can also converge quickly since the MLLM is already well-trained. For instance, under the setting of LLaVA Liu et al. (2023a), the full training process of RoE only takes about half the number of its default SFT tuning steps.

## 5 EXPERIMENT

### 5.1 DATASETS AND METRICS

The benchmarks used in this paper are five common vision-language and five recently proposed MLLM benchmarks. The common VL benchmarks include VQAv2 Goyal et al. (2017), GQA Hudson & Manning (2019), ScienceQA Lu et al. (2022), VizWiz Gurari et al. (2018) and TextVQA Singh

Table 1: Results of RoE with different skip rates on three MLLMs. "*Acc.*", "*Speed*" and "*Skip*" denote *accuracy*, *samples per second* and *the actual skip rate*, respectively.

| Method | SQA$^I$ | | | GQA | | | MMB | | | SEED | | | Average | | |
|---|---|---|---|---|---|---|---|---|---|---|---|---|---|---|---|
| | Acc. | Speed | Skip | Acc. | Speed | Skip | Acc. | Speed | Skip | Acc. | Speed | Skip | Acc. | Speed | Skip |
| LLaVA Liu et al. (2023a) | 66.8 | 7.55 | 0.00% | 62.0 | 6.99 | 0.00% | 64.3 | 8.37 | 0.00% | 58.6 | 8.33 | 0.00% | 62.9 | 7.81 | 0.00% |
| RoE-LLaVA$_{10\%}$ | 68.4 | 7.65 | 10.26% | 61.4 | 7.07 | 4.59% | 64.3 | 9.62 | 20.55% | 58.2 | 8.41 | 9.04% | 63.5 | 8.19 | 7.77% |
| RoE-LLaVA$_{20\%}$ | 68.7 | 9.15 | 20.55% | 61.3 | 7.52 | 7.86% | 64.6 | 9.88 | 23.64% | 57.8 | 9.85 | 24.52% | 63.1 | 9.10 | 19.15% |
| RoE-LLaVA$_{30\%}$ | 68.4 | 9.67 | 23.03% | 61.4 | 7.65 | 8.81% | 64.8 | 10.14 | 28.94% | 58.2 | 10.43 | 30.43% | 63.1 | 9.47 | 22.80% |
| VILA Lin et al. (2023) | 68.2 | 8.27 | 0.00% | 62.3 | 8.03 | 0.00% | 68.9 | 8.51 | 0.00% | 8.36 | 8.36 | 0.00% | 65.1 | 8.29 | 0.00% |
| RoE-VILA$_{10\%}$ | 69.5 | 8.39 | 9.19% | 62.2 | 8.01 | 4.83% | 67.6 | 8.63 | 10.59% | 61.3 | 8.50 | 11.41% | 65.2 | 8.38 | 11.94% |
| RoE-VILA$_{20\%}$ | 68.4 | 10.49 | 23.93% | 61.1 | 8.20 | 12.02% | 67.8 | 10.37 | 19.57% | 61.2 | 9.85 | 22.34% | 64.6 | 9.73 | 19.45% |
| RoE-VILA$_{30\%}$ | 69.4 | 10.67 | 25.12% | 60.3 | 8.21 | 13.41% | 66.8 | 11.66 | 27.56% | 60.2 | 10.73 | 27.66% | 64.2 | 10.32 | 23.44% |
| LLaVA-HR Luo et al. (2024c) | 65.1 | 4.82 | 0.00% | 64.2 | 4.87 | 0.00% | 64.9 | 4.76 | 0.00% | 64.2 | 3.74 | 0.00% | 64.6 | 4.55 | 0.00% |
| RoE-LLaVA-HR$_{10\%}$ | 67.4 | 4.96 | 7.96% | 62.5 | 5.01 | 7.65% | 64.6 | 4.82 | 6.96% | 62.2 | 3.86 | 8.43% | 64.2 | 4.66 | 7.68% |
| RoE-LLaVA-HR$_{20\%}$ | 56.1 | 4.97 | 12.77% | 60.8 | 5.09 | 11.07% | 52.9 | 4.89 | 10.63% | 58.8 | 3.92 | 13.62% | 57.2 | 4.72 | 12.02% |

Table 2: Ablation study of RoE. "*Acc.*", "*Speed*" and "*Skip*" denote *accuracy*, *samples per second* and *actual skip rate*, respectively. Here, "+*Reg*$_{20\%}$" refers to the use of the sparse regularization.

| Method | SQA$^I$ | | | GQA | | | MMB | | | SEED | | | Average | | |
|---|---|---|---|---|---|---|---|---|---|---|---|---|---|---|---|
| | Acc. | Speed | Skip | Acc. | Speed | Skip | Acc. | Speed | Skip | Acc. | Speed | Skip | Acc. | Speed | Skip |
| LLaVA | 66.8 | 7.55 | 0.00% | 62.0 | 6.99 | 0.00% | 64.3 | 8.37 | 0.00% | 58.6 | 8.33 | 0.00% | 62.9 | 7.81 | 0.00% |
| + Router | 69.0 | 7.37 | 3.23% | 61.2 | 6.29 | 0.01% | 65.5 | 7.64 | 0.03% | 58.4 | 7.67 | 1.13% | 63.5 | 7.24 | 1.10% |
| + Reg$_{20\%}$ | 64.3 | 8.63 | 15.48% | 59.6 | 7.57 | 7.00% | 63.8 | 9.32 | 17.89% | 56.6 | 9.18 | 18.49% | 61.1 | 8.59 | 14.72% |
| + Adapter | 68.7 | 9.15 | 20.55% | 61.3 | 7.52 | 7.86% | 64.6 | 9.88 | 23.64% | 57.8 | 9.85 | 24.52% | 63.1 | 9.10 | 19.15% |

Table 3: The training costs of RoE on three MLLMs. "*Adapter*" and "*Router*" denote the *adapter warmup* and *router warmup* of RoE. We use the GPU hour of A800 to measure the training time (*Time*). The values of "*Data*" are the number of examples for training. The base MLLMs only involve pretraining and SFT tuning ("*Finetune*").

| Method | Pretraining | | Adapter | | Router | | Finetune | | Total | |
|---|---|---|---|---|---|---|---|---|---|---|
| | Time | Data | Time | Data | Time | Data | Time | Data | Time | Data |
| VILA | 6326.5 | 50M | 0.0 | 0 | 0.0 | 0 | 120.9 | 1M | 6447.4 | 51M |
| **RoE-VILA** | 0.0 | 0 | 25.3 | 100k | 18.7 | 67k | 49.6 | 166k | 93.6 | 333k |
| LLaVA | 55.4 | 558k | 0.0 | 0 | 0.0 | 0 | 82.6 | 665k | 138.0 | 1.2M |
| **RoE-LLaVA** | 0.0 | 0 | 25.3 | 100k | 18.7 | 67k | 49.6 | 166k | 93.6 | 333k |
| LLaVA-HR | 37.2 | 558k | 0.0 | 0 | 0.0 | 0 | 128.4 | 665k | 165.6 | 1.2M |
| **RoE-LLaVA-HR** | 0.0 | 0 | 39.1 | 100k | 29.0 | 67k | 76.7 | 166k | 144.8 | 333k |

et al. (2019). During testing, we use the data splits organized in the instruction format of LLaVA-1.5 Liu et al. (2023a). And we report the accuracy of these datasets. The MLLM-specific benchmarks are POPE Li et al. (2023c), MME Fu et al. (2023), MMB Liu et al. (2023d), SEED Li et al. (2023a) and MM-Vet Yu et al. (2023). Compared to common VL evaluations, these benchmarks aim to evaluate MLLMs from various aspects, such as *fine-grained reasoning* and *visual hallucination*.

## 5.2 IMPLEMENTATION DETAILS

We apply RoE to three popular MLLMs, namely LLaVA-1.5 Liu et al. (2023a), LLaVA-HR Luo et al. (2024c) and VILA Lin et al. (2023), and term the new models as *RoE-LLaVA*, *RoE-LLaVA-HR* and *RoE-VILA*, respectively. For all MLLMs, the hidden dimension of RoE's adapter connections is set to 1,024. The hyper-parameter $\alpha$ is set to $0.5$ to control the impact of sparsity regularization. We randomly sample 15%, 10% and 25% of the *665k* instruction data of LLaVA-1.5 Liu et al. (2023a) for our three-stage training, respectively. During training, MLLMs are optimized with a learning rate of $2 \times 10^{-6}$, while the routers and adapter connections are updated with a learning rate of $4 \times 10^{-4}$. The training epoch is set to 1, and early stop is applied. The rest settings are kept the same with the original MLLMs. More details can refer to our code project.

Table 4: Comparison with existing MLLMs on 5 MLLM Benchmarks. *"Param."*, *"Res."*, *"Acc."* and *"Speed"* denote *parameter scale*, *input image resolution*, *accuracy* and *sample per second*, respectively. The best and second best results are marked in **bold** and underline, respectively.

| Method | LLM | Param. | Res. | POPE | | MME | | MMB | | SEED | | MM-Vet | |
|---|---|---|---|---|---|---|---|---|---|---|---|---|---|
| | | | | Acc. | Speed | Score | Speed | Acc. | Speed | Acc. | Speed | Score | Speed |
| *Dense MLLMs* | | | | | | | | | | | | | |
| Qwen-VL Bai et al. (2023b) | Qwen-7B | 9.6B | 448 | - | - | - | - | 38.2 | 7.40 | 56.3 | 2.42 | - | - |
| Qwen-VL-Chat Bai et al. (2023b) | Qwen-7B | 9.6B | 448 | - | - | 1487.5 | 3.96 | 60.6 | 7.55 | 58.2 | 2.59 | - | - |
| LLaVA Liu et al. (2023b) | Vicuna-7B | 7.2B | 336 | 85.9 | 8.90 | 1510.7 | 8.61 | 64.3 | 8.37 | 58.6 | 8.33 | 30.5 | 0.51 |
| VILA Lin et al. (2023) | Vicuna-7B | 7.2B | 336 | 85.5 | 9.21 | 1533.0 | 8.64 | **68.9** | 8.51 | 61.1 | 8.36 | 34.9 | 0.48 |
| LLaVA-HR Luo et al. (2024c) | Vicuna-7B | 7.4B | 1024 | 85.9 | 4.70 | 1554.9 | 4.77 | 64.9 | 4.48 | **64.2** | 3.46 | 31.2 | **0.76** |
| *Sparse MLLMs* | | | | | | | | | | | | | |
| MoE-LLaVA-1.6B×4 Lin et al. (2024) | StableLM-1.6B | 2.9B | 336 | 85.7 | 7.65 | 1318.2 | 8.06 | 60.2 | 9.90 | - | - | 26.9 | 0.43 |
| MoE-LLaVA-2.7B×4 Lin et al. (2024) | Phi-2.7B | 5.3B | 336 | 86.3 | 5.95 | 1423.0 | 5.83 | 65.2 | 5.27 | - | - | 34.3 | 0.25 |
| **RoE-LLaVA** | Vicuna-7B | 7.3B | 336 | 86.1 | **9.38** | 1522.7 | **9.03** | 64.3 | **9.62** | 58.2 | 8.41 | 31.9 | 0.42 |
| **RoE-VILA** | Vicuna-7B | 7.3B | 336 | 86.8 | 9.25 | 1446.0 | 8.95 | 67.6 | 8.63 | 61.3 | **8.50** | 36.7 | 0.43 |
| **RoE-LLaVA-HR** | Vicuna-7B | 7.5B | 1024 | **88.1** | 4.75 | **1558.2** | 4.82 | 64.6 | 4.82 | 62.2 | 3.86 | 30.0 | 0.68 |

Table 5: Comparison with existing MLLMs on 5 traditional VL benchmarks. *"Param."*, *"Res."*, *"Acc."* and *"Speed"* denote *parameter scale*, *input image resolution*, *accuracy* and *sample per second*, respectively. The best and second best results are marked in **bold** and underline, respectively.

| Method | LLM | Param. | Res. | VQA$^{v2}$ | | GQA | | VizWiz | | SQA$^I$ | | VQA$^T$ | | Average | |
|---|---|---|---|---|---|---|---|---|---|---|---|---|---|---|---|
| | | | | Acc. | Speed | Acc. | Speed | Acc. | Speed | Acc. | Speed | Acc. | Speed | Acc. | Speed |
| *Dense MLLMs* | | | | | | | | | | | | | | | |
| Qwen-VL Bai et al. (2023b) | Qwen-7B | 9.6B | 448 | 78.8 | 5.23 | 59.3 | 3.48 | 35.2 | 3.92 | 67.1 | 6.97 | 63.8 | 3.77 | 60.8 | 4.67 |
| Qwen-VL-Chat Bai et al. (2023b) | Qwen-7B | 9.6B | 448 | 78.2 | 5.30 | 57.5 | 3.63 | 38.9 | 3.22s | 68.2 | 6.10 | 61.5 | 5.21 | 60.9 | 4.69 |
| LLaVA Liu et al. (2023b) | Vicuna-7B | 7.2B | 336 | 78.5 | 6.97 | 62.0 | 6.99 | 50.0 | 6.44 | 66.8 | 7.55 | 58.2 | **5.84** | 63.1 | 6.76 |
| VILA Lin et al. (2023) | Vicuna-7B | 7.2B | 336 | 79.9 | 8.01 | 62.3 | **8.03** | **57.8** | 5.75 | 68.2 | 8.27 | 64.4 | 5.70 | **65.5** | 7.15 |
| LLaVA-HR Luo et al. (2024c) | Vicuna-7B | 7.4B | 1024 | **81.9** | 4.42 | **64.2** | 4.55 | 48.7 | 4.06 | 65.1 | 4.71 | **67.1** | 3.81 | 65.4 | 4.31 |
| *Sparse MLLMs* | | | | | | | | | | | | | | | |
| MoE-LLaVA-1.6B×4 Lin et al. (2024) | StableLM-1.6B | 2.9B | 336 | 76.7 | 7.79 | 60.3 | 7.43 | 36.2 | 6.27 | 62.6 | 8.09 | 50.1 | 4.48 | 57.2 | 6.81 |
| MoE-LLaVA-2.7B×4 Lin et al. (2024) | Phi-2.7B | 5.3B | 336 | 77.6 | 6.01 | 61.4 | 5.23 | 43.9 | 3.95 | 68.5 | 5.80 | 51.4 | 3.76 | 60.6 | 4.95 |
| **RoE-LLaVA** | Vicuna-7B | 7.3B | 336 | 80.3 | 7.02 | 61.4 | 7.07 | 52.5 | **6.52** | 68.4 | 7.65 | 56.8 | 5.59 | 63.8 | 6.77 |
| **RoE-VILA** | Vicuna-7B | 7.3B | 336 | 78.8 | **8.25** | 62.2 | 8.01 | 53.7 | 6.28 | **69.5** | 8.39 | 59.3 | 5.75 | 64.7 | **7.34** |
| **RoE-LLaVA-HR** | Vicuna-7B | 7.5B | 1024 | 80.9 | 4.79 | 62.5 | 5.01 | 47.6 | 4.12 | 67.4 | 4.96 | 64.6 | 4.02 | 64.6 | 4.58 |

## 5.3 EXPERIMENTAL RESULTS

### 5.3.1 QUANTITATIVE ANALYSIS

**Results of RoE on different MLLMs.** In Table. 1, we first present the results of RoE applying to LLaVA-1.5 Liu et al. (2023a), LLaVA-HR Luo et al. (2024c) and VILA Lin et al. (2023) with different skip rates. From this table, we can observe that RoE can help existing MLLMs to reduce a large number of redundant parameters and computations. For instance, RoE-VILA$_{30\%}$ skips 23.44% of the parameters and speed up inference by about 24.5% on average. Notably, the performance of RoE-VILA$_{30\%}$ only drops by 1.38%. We can also observe that the impact of skip rate is distinct for different MLLMs. Specifically, for RoE-LLaVA, the performance does not drop obviously as the skip rate increases, *i.e.* -0.2% on average with an actual skip rate of 22.80%, while LLaVA-HR is more sensitive to network skipping. For example, the increase of skip rate from 7.68% to 12.02% results in about -4.34% performance drop on average. Nevertheless, RoE can further improve the compactness of LLaVA-HR through its dynamic routing. Another observation from these results is that the RoE-MLLM always has a higher skip rate on multiple-choice questions than the open-ended ones, *e.g.* 23.03% on SQA *vs.* 8.81% GQA on RoE-LLaVA$_{30\%}$, suggesting its actual skip rate is related to the task difficulty. Overall, these results well validate the effectiveness of our RoE and also confirm the redundancy of existing MLLMs.

**Ablation Study.** In Tab 2, we conduct a set of experiments to ablate the designs of RoE. In this table, "*+Router*" means that we directly equip LLaVA-1.5 with path routers as described in Eq.2. We can first observe that this direct attempt is hard to achieve the expected structure sparsity on LLaVA. As shown in Tab 2, its average skip rate is only 1.1%, and it is even as low as 0.1% on the difficult tasks like GQA. This result suggests that the model barely chooses to skip layers during inference, which well confirms our assumption about the challenges of routing the well-trained MLLMs. In practice, the additional computation of routers even leads to the latency of +7.29%. Based on "*+Router*", "*+Reg$_{20\%}$*" further applies the structure sparsity regularization to optimize the routers. With this regularization term, routers are better trained to evaluate and skip the less important layers of MLLMs. For instance, RoE-LLaVA can skip up to 10 layers on MMB, greatly improving the inference speed by +21.98%. Nevertheless, we still observe obvious performance

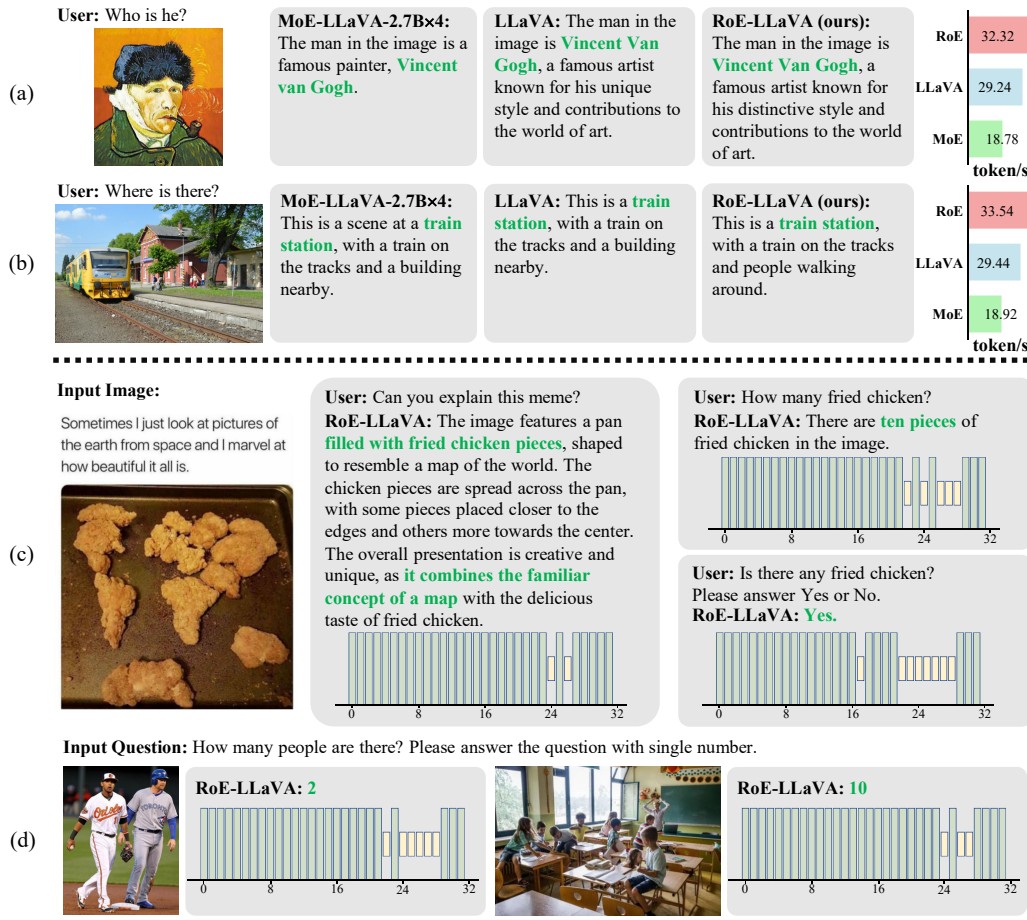

Figure 3: Examples of our RoE on LLaVA. Example (a) and (b) show the comparison between RoE-LLaVA and LLaVA and MoE-LLaVA. Our RoE-LLaVA can answer the questions as accurately and in detail as LLaVA, while being faster, *i.e.*, more *tokens per second* (**tokens/s↑**). Example (c) shows the predictions of RoE-LLaVA for the same image and different questions. RoE can adjust the choice and depth of expert pathways according to the question, *i.e.*, the bar charts (the yellow ones denote the skipped layers). Example (d) shows the predictions for the same question but different images. RoE can also route different optimal expert pathways according to different visual content.

degradation after layer skipping, *e.g.,* -2.0% on SEED. As we discussed in Sec. 4.1, directly skipping layers typically leads to dramatic changes in feature space. To compensate for this, the lightweight adapter-based skip connections are then ablated, *i.e.*, "*+Adapter*" in Tab 2. It can be seen that this simple design can greatly improve the average performance by up to +2.0%. Overall, these ablation results well confirm our motivation for dynamic routing in the well-trained MLLMs, and also validate the designs of RoE.

**The training costs of RoE.** In Tab.3, we report the training costs of RoE on three base MLLMs. From this table, we can observe that the implementation of RoE is much cheaper than building a new MLLM. For instance, training VILA from scratch takes about 6447.4 GPU hours, but its extension to VILA-RoE is only about 93.6 GPU hours since the pre-training is not required. In addition, the optimization of RoE is also very efficient. We can see that for all three base MLLMs, our RoE not only requires fewer SFT examples to train, but also has a shorter total training time than the SFT tuning of these models. Overall, from these statistics, we can conclude that RoE is also a training-efficient method for the improvement of existing MLLMs.

**Comparison with existing MLLMs**. In Tab. 4 and Tab. 5, we compare the performance and efficiency of RoE-MLLMs with more MLLMs. In Tab. 4, we can first observe the comprehensive advantages of RoE-MLLMs over the compared sparse MLLMs on four MLLM-specific benchmarks. In particular, the computation of MoE-LLaVA series is more expensive than our RoE-VILA although its parameter scale is smaller. For instance, compared with MoE-LLaVA-1.6×4, RoE-

VILA improves the scores by 9.8% on MM-Vet. And RoE-VILA also improves the inference speed by 1.11 times on MME. Similar advantages of RoE can also be witnessed in PoPE. Compared to MoE-LLaVA-2.7B×4, RoE-VILA not only achieves +0.5% performance gains but also speeds up the inference by 55.5%. When compared to the dense MLLMs, the benefits of RoE-MLLMs are still obvious. For instance, RoE-LLaVA-HR improves the score by +3.3 on MME, and RoE-VILA achieves +1.8 performance gains on MM-Vet, while still keeping a faster inference speed. Tab. 5 gives the performance comparison on common VL tasks. Compared to other sparse MLLMs, RoE-MLLMs still achieve better results on all benchmarks with superior efficiency. For instance, RoE-LLaVA outperforms MoE-LLaVA-2.7B×4 by +2.7 on VQA, while being +16.8% faster. Compared to dense MLLMs, the proposed RoE-MLLMs also show distinct advantages in efficiency, which can speed up the inference by 2.65%-6.26%. In terms of performance, RoE-MLLMs even outperform the original dense MLLMs on several benchmarks, *e.g.*, +1.6 of RoE-LLaVA against LLaVA on ScienceQA. Overall, these results further confirm the great effectiveness and efficiency of our RoE.

### 5.3.2 QUALITATIVE ANALYSIS

To gain insight into the proposed RoE, we visualize its predictions and skipped layers in Fig. 3. In Fig. 3 (a)-(b), we first compare its predictions with LLaVA and MoE-LLaVA-2.7B×4. We can observe that the implement of RoE barely impedes the expression compared with the default LLaVA. As shown in Fig.3 (a), RoE-LLaVA can give accurate and detailed answers for the question like the default LLaVA. In contrast, the response of the other sparse model MoE-LLaVA is briefer. Besides, it can be also seen that RoE-LLaVA has a much faster inference speed than the compared models for the same question. For two examples, RoE-LLaVA can speed up inference by up to 13.9% and 77.3% than LLaVA and MoE-LLaVA, respectively. In Fig. 3 (c)-(d), we examine the behaviors of RoE in terms of the image and the question, respectively. In Fig. 3 (c), RoE-LLaVA is required to answer different questions for the same image. From these examples, we can see that RoE skips fewer layers when answering the first question, which requires a detailed answer. In contrast, for the other two questions of which answer are simpler, RoE can skip more layers during inference. These results suggest that the actual skip rate of RoE is related to the answer content, and the more complex the answer, the fewer layers the RoE skips. In Fig. 3 (d) shows the response of RoE-LLaVA under the same question but different images. For the first image, of which scene is simpler, RoE can skip more layers to answer the number of people. Instead, RoE requires more reasoning steps to infer the answer of the second example, where the classroom scene is more complex. In this case, the behavior of RoE is also related to the complexity of visual content. Overall, from these examples, we can conclude that RoE is example-dependent and can route the optimal expert paths for different examples according to their difficulties, which well confirms our motivation.

## 6 CONCLUSION

In this paper, we focus on exploring the dynamic experts in existing MLLMs, and propose a novel and effective modeling scheme called ***Routing Expert*** (RoE). To address the challenges for the well-trained MLLMs, RoE introduces the adapter-based skip connection to mitigate the issue of feature gap and a novel sparsity regularization to encourage MLLMs to learn sparse inference. Meanwhile, the routing token design is also proposed to address the training and testing misalignment of existing MLLMs. Extensive experiments on 3 MLLMs and 10 benchmarks demonstrate that our RoE method can greatly improve the efficiency of existing MLLMs without obvious structure modification and with low training cost, while maintaining close or even better performance.

## 7 ACKNOWLEDGE

This work was supported by the National Science Fund for Distinguished Young Scholars (No.62025603), the National Natural Science Foundation of China (No. U21B2037, No. U22B2051, No. U23A20383, No. U21A20472, No. 62176222, No. 62176223, No. 62176226, No. 62072386, No. 62072387, No. 62072389, No. 62002305 and No. 62272401), the Natural Science Foundation of Fujian Province of China (No. 2021J06003, No.2022J06001) and the Fundamental Research Funds for the Central Universities (Xiamen University: No. 20720240053).

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
