# OpenReview forum: "Routing Experts: Learning to Route Dynamic Experts in Existing Multi-modal Large Language Models"
_ICLR.cc/2025/Conference — ICLR 2025 Poster_

### Official Review · Reviewer_2K7T · 2024-11-04

**Soundness:** 3
**Presentation:** 3
**Contribution:** 3
**Rating:** 6
**Confidence:** 3

**Summary:**

This paper introduces Routing Experts (RoE), a novel approach to transform existing multimodal LLMs into mixture-of-experts models without significant architectural changes. The key innovation is treating each layer of pre-trained MLLMs as a potential expert that can be dynamically routed or skipped based on input complexity.

**Strengths:**

- The paper is well-written and good in coherence.
- The authors present innovative architectural designs that effectively integrate Mixture-of-Depths into Multimodal Large Language Models
- The empirical validation is comprehensive, encompassing diverse model architectures and benchmark datasets.
- The proposed approach is computationally efficient, requiring minimal fine-tuning overhead for implementation.

**Weaknesses:**

**Major**
- From Table 1, RoE-LLaVA-HR shows a large drop in performance. While the authors note that "LLaVA-HR is more sensitive to network skipping ... Nevertheless, RoE can further improve the compactness." They should explain why this happens and whether the improved compactness is worth the performance loss.
- From Table 2, comparing RoE to *Router* that entirely skips model layers may not be fair enough. The study needs separate tests for each part of RoE (adapter, regularization, and router token) to show how each contributes.
- The sparsity ratio in Table 4 and 5 is not clearly stated, and the inference speed improvements are not very impressive. This raises questions about how well RoE can handle more complex tasks and higher sparsity levels.

**Minor**
- Formatting: Too much empty space under figures and between sections.
- Inconsistent Terms: "L1-Distance" is written differently in Figure 1 and its caption.

**Questions:**

- While the study demonstrates thorough experiments across different LLaMA-based MLLMs, the generalizability to non-LLaMA architectures (e.g., Qwen) remains unexplored. Testing RoE on diverse language model backbones would better validate its broader applicability.

---

> ### Author Response · Authors · 2024-11-23
>
> # Comment to Reviewer 2K7T
>
> We highly appreciate your time and effort in reviewing this paper, as well as your encouraging and constructive comments on our work. Below, we response to your key concerns point by point.
>
>
> **Comment 1:** From Table 1, RoE-LLaVA-HR shows a large drop in performance. While the authors note that "LLaVA-HR is more sensitive to network skipping ... Nevertheless, RoE can further improve the compactness." They should explain why this happens and whether the improved compactness is worth the performance loss.
>
> **Response:** Thanks for your comments.
> Due to a more complex visual feature learning scheme, the joint optimization of LLaVA-HR is more difficult than other LLaVA-like MLLMs.
> In this case, without changing the hyper-parameters, RoE achieves less satisfactory results, as reported in the paper.
>
> We note that it is due to insufficient router warm-up.
> In the following experiments, we enhance the router warmup stage with an additional 5% of the data from the SFT dataset in LLaVA, and the obtained results are better, as shown below.
>
> We will update these results in our paper.
>
> |Method|SQA Acc|SQA Speed|SQA Skip|SQA TFLOPs|GQA Acc|GQA Speed|GQA Skip|GQA TFLOPs|
> |-|-|-|-|-|-|-|-|-|
> |LLaVA-HR|65.1|4.82|0.00%|18.2|64.2|4.87|0.00%|17.3|
> |RoE-LLaVA-HR_{20%}|68.2|4.89|4.86%|17.4|62.6|5.03|8.20%|16.1|
>
> **Comment 2:** From Table 2, comparing RoE to Router that entirely skips model layers may not be fair enough. The study needs separate tests for each part of RoE (adapter, regularization, and router token) to show how each contributes.
>
> **Response:** Thanks for this comment.
> Considering that the target of RoE is to transform MLLMs into dynamic models for better efficiency, so we use MLLMs+router as our baseline in Tab.2.
> Without routers, MLLM+adpater is a static and dense model.
> In the third row of Tab.2, *i.e.*, +Reg20%, shows the effects of sparsity regularization, which help the model increase skip rate but with obvious performance drop.
>
> Following your suggestion, we supplement the results of router+adapter below.
>
> |Method|SQA Acc|SQA Speed|SQA Skip|GQA Acc|GQA Speed|GQA Skip|
> |-|-|-|-|-|-|-|
> |LLaVA|66.8|7.55|0.00%|62.0|6.99|0.00%|
> |+Router+Adapter|68.8|7.44|4.45%|61.3|6.31|0.02%|
> |RoE-LLaVA|68.7|9.15|20.55%|61.3|7.52|7.86%|
>
> It can be seen that this setting will not lead to obvious performance loss, but the actual skip rate is still limited.
> To further address your concern, we also report the results of the manual setting of skipping layers below, *i.e.*, directly using adapter to skip fixed layers.
> We select several layers with the highest exit probability in RoE as fixed exit layers.
>
> |Benchmark|RoE|Skip_{2}|Skip_{4}|Skip_{6}|
> |-|-|-|-|-|
> |GQA|61.4|60.6|58.7|57.1|
> |TextVQA|56.8|54.7|51.2|46.1|
>
> We can see that this static method also leads to obvious decline in performance, suggesting the need for dynamic routing by RoE.
> Overall, the best results are the combination of three designs in the last row of Tab.2, *i.e.*, +adpater.
>
> **Comment 3:** The sparsity ratio in Table 4 and 5 is not clearly stated, and the inference speed improvements are not very impressive. This raises questions about how well RoE can handle more complex tasks and higher sparsity levels.
>
> **Response:** Thanks for this careful review.
> The target skip ratios in tab.4 and 5 are set to 10%.
>
> In terms of your last question, we think that RoE is not expected to achieve very high sparsity for more complex tasks.
>
> To explain, one main motivation of RoE is that simple examples do not require such a dense MLLM with billions of parameters to answer the question, which is obviously redundant.
> Moreover, simple examples often make up the major use of MLLMs in practical applications, akin to LLMs.
> Thus, dense inference will waste massive computation overhead. To this end, we can see that RoE can achieve very high sparsity in tasks like SQA [2] and MMB [3], with easier instructions.
>
> However, for difficult examples, especially those reaching the capability upper limit of MLLMs, we think that RoE should account for fewer sparsity for better reasoning in MLLMs in contrast. This assumption is similar to the finding of recent OpenAI o1, which shows that MLLMs need more reasoning steps to handle difficult tasks.
>
> [2] Lu, Pan, et al. "Learn to explain: Multimodal reasoning via thought chains for science question answering" NeurIPS 2022
>
> [3] Liu, Yuan, et al. "Mmbench: Is your multi-modal model an all-around player?" ECCV 2025
>
> **Comment 4:** Formatting: Too much empty space under figures and between sections.
>
> **Response:** Thanks for your careful review, and we will keep on polishing the paper until the final submission.
>
> **Comment 5:** Inconsistent Terms: "L1-Distance" is written differently in Figure 1 and its caption.
>
> **Response:** Thanks for pointing out this typo, and we will revise the consistency of this terminology.

---

> > ### Author Response · Authors · 2024-11-23
> >
> > **Comment 6:** While the study demonstrates thorough experiments across different LLaMA-based MLLMs, the generalizability to non-LLaMA architectures (e.g., Qwen) remains unexplored. Testing RoE on diverse language model backbones would better validate its broader applicability.
> >
> > **Response:** Thanks for this constructive suggestion. We acknowledge that QWen series are a very outstanding and representative MLLM/LLM family with a wide impact on both academia and industry.
> > But due to the rebuttal time limit, we fail to reproduce the training code project for VL tasks, and we will leave it to our near future work. Besides, we will actively cite and discuss QWen series in our final submission.
> >
> > In addition, we apply RoE to the LLaMA-3 8B, and report the results in the following table.
> >
> > |Method|Skip Ratio|Commonsense QA|Skip Ratio|Logiqa|Skip Ratio|WNLI|
> > |-|-|-|-|-|-|-|
> > |LLaMA-3 8B|0.0%|76.25|0.0%|27.65|0.0%|61.97|
> > |RoE_30%|15.8%|75.10|6.42%|29.03|3.83%|60.56|
> >
> > It can be seen that our RoE is also applicable to representative LLMs without careful tuning.
> > We think this application will be more prominent on larger LLMs, but due to the time limit, we leave it in our near future work.

---

> ### Author Response · Authors · 2024-11-25
>
> Dear Reviewer 2K7T,
>
> Thanks again for your great efforts and constructive advice in reviewing this paper! With the discussion period drawing to a close, we expect your feedback and thoughts on our reply. We put a significant effort into our response, with several new experiments and discussions. We sincerely hope you can consider our reply in your assessment.
>
> We look forward to hearing from you, and we can further address unclear explanations and remaining concerns if any.
>
> Regards,
>
> Authors

---

> ### Author Response · Authors · 2024-11-28
> **Looking forward to the feedback from Reviewer 2K7T**
>
> Dear Reviewer 2K7T,
>
> We hope that our detailed rebuttal can address your concerns about this paper.
>
> As the deadline is approaching, we are looking forward to your valuable feedback and also welcome any new questions you may have.
>
> Thanks again for your time and efforts in reviewing this paper.
>
> Best regards
>
> The authors

---

> > ### Comment · Reviewer_2K7T · 2024-11-29
> >
> > Thanks for the detailed explanation. I will raise my score a bit.

---

> > > ### Author Response · Authors · 2024-11-29
> > > **Official Comment by Authors**
> > >
> > > Thanks a lot for your valuable feedback and constructive review. We highly appreciate it.

---

### Official Review · Reviewer_akxJ · 2024-11-04

**Soundness:** 3
**Presentation:** 2
**Contribution:** 2
**Rating:** 6
**Confidence:** 3

**Summary:**

This paper explores how to make a MLLM (multimodal large language model) more efficient. They find that different layers of the MLLM contribute differently to each sample; therefore, the paper proposes adaptively skipping over less important layers. They do this by replacing full layers with adapters, and learning a routing function at each layer that makes a binary decision to use the adapter or the full layer. This effectively converts an existing MLLM into a "mixture of experts". They find that this approach, called RoE, results in significant speedups at inference time while suffering only a slight degradation in accuracy.

**Strengths:**

Originality: simple way of taking an existing model and making it more efficient/into a MOE by swapping out existing layers with adapters and training a router.

Quality: results show fairly consistent speedups in MLLMs. RoE also outperforms existing MOEs in accuracy on downstream tasks.

**Weaknesses:**

Quality:
- Only one scenario (SQA in table 5) has RoE being strictly better than other models in both accuracy and speed. All other settings exhibit a tradeoff, and it is not unclear how good/bad this tradeoff is. It would be nice if the paper could visualize the pareto frontier. There are also some cases where RoE is slower than dense MLLM counterparts.
- Some additional ablations would be helpful, such as adapter+no router+no reg; that is, just using the adapter at each layer.
- More analysis would be better, i.e., what is being routed between the adapter and the existing layer (Figure 3d touches on this)

Clarity:
- Notation like $\{G_1, G_2, \dots, G_n\}$ are layers chosen by the router. Its number is smaller than the default length $n$ is ambiguous. Should use separate $n$'s.
- Equation (3) is not properly explained - what is $I$? What is $T$?
- What are the training objectives for Stage 1 and Stage 2 of RoE?

Significance: there are many tradeoffs in the proposed method. Therefore, it is unclear how much people would use this in practice (i.e., do people want to spend ~93.6 GPU hours to make an existing model faster but worse---and not clear by how much)

**Questions:**

1. Can you provide more visualization on the accuracy/speed tradeoff of each model?
2. Can you provide additional ablations, such as showing what happens when a subset of the layers is deterministically set to use the adapter.
3. Can you provide more examples that are routed to adapters more often versus routed to the existing layer more often? Do these correspond with easier/harder samples?
4. Clarity suggestions.
5. Would welcome more discussion/results on improving and quantifying the tradeoffs made in RoE and its costs.

---

> ### Author Response · Authors · 2024-11-23
>
> # Comment to Reviewer akxJ
>
> We highly appreciate your time and effort in reviewing this paper. Below, we response to your key concerns point by point.
>
> **Comment 1:** Only one scenario (SQA in table 5) has RoE being strictly better than other models in both accuracy and speed. All other settings exhibit a tradeoff, and it is not unclear how good/bad this tradeoff is. It would be nice if the paper could visualize the pareto frontier. There are also some cases where RoE is slower than dense MLLM counterparts.
>
> **Response:** Thanks for this comment.
> Following your suggestion, we visualize the Pareto frontier of Tab.1 in our anonymous project: https://anonymous.4open.science/r/RoE_Rebuttal-5A77/pareto.png.
> From this plot, we can observe that the proposed RoE can achieve well trade-off between performance and efficiency.
>
> **Comment 2:** Some additional ablations would be helpful, such as adapter+no router+no reg; that is, just using the adapter at each layer.
>
> **Response:** Thanks for your suggestion.
> In tab.2, we ablate the effects of router, sparsity regularization and adapter. In particular, your mentioned setting of \`\`adapter+no router+ no reg'' is a dense network setting, *i.e.*, no layers will be skipped, which is beyond the motivation and target of this paper.
>
> **Comment 3:** More analysis would be better, i.e., what is being routed between the adapter and the existing layer (Figure 3d touches on this)
>
> **Response:** Thanks for this insightful question.
> In terms of quantitative analysis, the actual routing rate of RoE depends on the task difficulty as shown in Tab.1.
> For instance, on benchmarks such as SQA and MMB, which are mainly about simple multi-choice questions, RoE-LLaVA can skip up to 30.43% of layers of MLLMs on average.
> In contrast, tasks like GQA require granular multimodal reasoning, MLLMs-RoE often requires more layers to answer the question.
>
> In terms of qualitative analysis, the routing of RoE is related to instruction difficulty (or answer difficulty) and visual complexity, as shown in Fig.3. Simple put, if the image and the instruction are simple and common, RoE can help MLLMs skip more layers for efficient inference with accurate answering, and vice verse.
>
> Overall, these results show that RoE can help MLLMs skip more layers to answer simple instructions, which is also akin to the recent findings of OpenAI o1 that a difficult task needs more reasoning steps in MLLMs.
>
> We will follow your suggestion to supplement more discussions to our new submission.
>
> **Comment 4:** Notation like G_1, G_2, G_3, ..., G_n are layers chosen by the router. Its number is smaller than the default length n is ambiguous. Should use separate n's.
>
> **Response** Thanks for your careful review.
> We will revise these typos and keep on polishing our paper until the final submission.
>
> **Comment 5:** quation (3) is not properly explained - what is I ? What is T ?
>
> **Response:** Thank you for your question.
> In Eq.3, \`\`I'' denotes the input image and \`\`T'' denotes the input text.
>
> **Comment 6:** What are the training objectives for Stage 1 and Stage 2 of RoE?
>
> **Response:** Thanks for this comment.
> As introduced in Sec. 4.4, we only use the default SFT objective of MLLMs to train adapters at the first stage, while the sparsity regularization is then used in the last two stages.
>
> **Comment 7:** There are many tradeoffs in the proposed method. Therefore, it is unclear how much people would use this in practice (i.e., do people want to spend ~93.6 GPU hours to make an existing model faster but worse---and not clear by how much)
>
> **Response:** Thanks for this comment.
> In this paper, we study the redundancy of MLLMs from a new perspective in addition to the popular MoE, and also propose a viable and efficient paradigm RoE.
>
> As discussed above, RoE can help MLLMs skip more layers for better efficiency on simple tasks.
> For instance, RoE-LLaVA reduces about 26.2% FLOPs on SQA without performance drops.
> Notably, these simple instructions often make up the majority during practical use.
> Existing online MLLM systems also suffer excessive computation from these numerous simple user requests.
> To this end, RoE is of great significance for the efficient application of MLLMs.
>
> Meanwhile, as commented by other reviewers, our work \`\`solved an interesting and meaningful problem'' (R#PWB7), \`\`shows good seedups without much accuracy degradation'' (R#iEWz) and \`\`presents innovative architectural designs'' (R#2K7T).
> Thus, we believe that our RoE can well facilitate the development of MLLMs.

---

> > ### Author Response · Authors · 2024-11-23
> >
> > **Comment 8:** Can you provide additional ablations, such as showing what happens when a subset of the layers is deterministically set to use the adapter.
> >
> > **Response:** Thanks for your comment.
> > Following your suggestion, we report the performance of RoE under your suggested fix-layer setting, as shown below.
> > It can be seen that direct manually skipping a fixed layers of LLaVA will leads to a drastic decline in performance, suggesting the need of example-dependent routing.
> >
> > |Benchmark|RoE|Skip_{2}|Skip_{4}|Skip_{6}|
> > |-|-|-|-|-|
> > |GQA|61.4|60.6|58.7|57.1|
> > |TextVQA|56.8|54.7|51.2|46.1|
> >
> > **Comment 9:** Can you provide more examples that are routed to adapters more often versus routed to the existing layer more often? Do these correspond with easier/harder samples?
> >
> > **Response:** Following your suggestion, we visualize more examples in our anonymous project: https://anonymous.4open.science/r/RoE_Rebuttal-5A77/visualization.png.
> > From the visualizations, we can observe that the number of activated layers is related to the difficulty of the given questions.
> > For instance, as shown in the left column, when the problem is only related to the image content and only requires simple reasoning, the model tends to have a high sparsity.
> > And the complex image also limits the sparsity.
> > On the other hand, problems involving more external knowledge, such as Mona Lisa, will lead to fewer layers being skipped in the model.
> > The complex reasoning, *e.g.*, asking the reason for the boy's wet clothes, also leads to fewer skipped layers.
> >
> > In addition, we can also see some additional patterns of RoE.
> > For instance, the base layers of MLLMs are often less skipped, suggesting the importance of shallow multimodal fusion.
> > In contrast, the high-level layers often show more obvious redundancy.
> > These findings are also consistent with recent efficient LLM researches [2,3].
> >
> > [1] Saikh, Tanik, et al. "ScienceQA: A Novel Resource for Question Answering on Scholarly Articles" Int. J. Digit. Libr. 2022
> >
> > [2] Chen, Yanxi, et al. "Ee-llm: Large-scale training and inference of early-exit large language models with 3d parallelism" arXiv preprint arXiv:2312.04916 (2023)
> >
> > [3] Wu, Qiong, et al. "Parameter and computation efficient transfer learning for vision-language pre-trained models" Advances in Neural Information Processing Systems 2024
> >
> > **Comment 10:** Would welcome more discussion/results on improving and quantifying the tradeoffs made in RoE and its costs.
> >
> > **Response:** Thanks for your suggestion. We expect two main advantages and a favorable trade-off in RoE's performance and cost:
> >
> > On larger MLLMs, the benefit of skipping layers is more pronounced. As each layer has more parameters, the efficiency gains from skipping are more significant. Additionally, the stronger ability of each layer in larger MLLMs allows for higher sparsity, especially in simpler tasks.
> >
> > With more SFT data, RoE can learn more accurate skip strategies across various tasks. It further enhances sparsity and reduces unnecessary computation. The diversity of tasks helps RoE refine its routing decisions, making it more adaptable.
> >
> > Besides, RoE can keep its low training cost on larger MLLMs. Since RoE extends a pre-trained dense model, there is no need for an extensive pre-training phase when applying RoE to larger models. This helps maintain a lower computational cost compared to training from scratch, even when scaling up.

---

> ### Author Response · Authors · 2024-11-25
>
> Dear Reviewer akxJ,
>
> Thanks again for your great efforts and constructive advice in reviewing this paper! With the discussion period drawing to a close, we expect your feedback and thoughts on our reply. We put a significant effort into our response, with several new experiments and discussions. We sincerely hope you can consider our reply in your assessment.
>
> We look forward to hearing from you, and we can further address unclear explanations and remaining concerns if any.
>
> Regards,
>
> Authors

---

> ### Author Response · Authors · 2024-11-28
> **Looking forward to the feedback from Reviewer akxj**
>
> Dear Reviewer akxJ,
>
> We hope that our detailed rebuttal can address your concerns about this paper.
>
> As the deadline is approaching, we are looking forward to your valuable feedback and also welcome any new questions you may have.
>
> Thanks again for your time and efforts in reviewing this paper.
>
> Best regards
>
> The authors

---

### Official Review · Reviewer_iEwZ · 2024-11-06

**Soundness:** 4
**Presentation:** 4
**Contribution:** 3
**Rating:** 8
**Confidence:** 4

**Summary:**

This paper introduces a method of introducing sparsity in multimodal LLMs by skipping some transformer layers. Importantly the skipped layers have a low-rank adaptor applied, and the exact set of layers skipped or not depends on the input due to a learned routing function. The work is in essence combining ideas of model pruning with MoEs and applying it to multimodal LLMs. The authors introduce several techniques to effectively train this model, such as warming up the routers and adaptors, and enforcing sparsity in the router. They show their model can maintain very close to SOTA accuracy across a variety of multimodal LLMs while increasing throughput by 10-20%.

**Strengths:**

The paper is clearly written, and the main ideas are simply presented. The work shows good speedups of up to 20% without much degradation of model accuracy.

**Weaknesses:**

The paper focuses mostly on the connection of their work to MoEs, but not as much on the connection to existing model pruning / layer removal efforts. Also while the paper compares accuracy & speed-up compared to the baseline models, they don't compare to baseline pruning or distillation techniques.

**Questions:**

Suggestion for improvement is to compare against another technique for model pruning or distillation.

---

> ### Author Response · Authors · 2024-11-23
>
> # Comment to Reviewer iEwZ
>
> We highly appreciate your time and effort in reviewing this paper, and also thanks a lot for your constructive and encouraging comments on our work.
>
> **Comment 1:** The paper focuses mostly on the connection of their work to MoEs, but not as much on the connection to existing model pruning / layer removal efforts.
> Also while the paper compares accuracy & speed-up compared to the baseline models, they don't compare to baseline pruning or distillation techniques.
>
> **Response:** Thanks for your comment.
> RoE mainly considers the layer-wise redundancy in exiting MLLMs, of which contribution is orthogonal to the above methods.
> better respond your concern, we provide the experiment results of combine or RoE with token pruning method Fastv as follows:
>
> |Mtehod|SQA Acc|SQA Skip Ratio|SQA Speed|SQA TFLOPs|GQA Acc|GQA Skip Ratio|GQA Speed|GQA TFLOPs|
> |-|-|-|-|-|-|-|-|-|
> |LLaVA|66.8|0.00%|7.55|9.8|62.0|0.00%|6.99|8.8|
> |RoE-LLaVA|68.4|23.03%|9.67|7.2|61.4|8.81%|7.65|7.7|
> |RoE-LLaVA+Fastv|69.16|21.03%|12.1|4.6|60.0|7.98%|9.34|5.0|
>
> It can be seen that our RoE is fully compatible with advanced token pruning methods like FastV, well confirming its generalization and robustness.

---

### Official Review · Reviewer_PWB7 · 2024-11-08

**Soundness:** 3
**Presentation:** 3
**Contribution:** 2
**Rating:** 6
**Confidence:** 4

**Summary:**

This paper proposes RoE, which skips layers in an existing MLLM achieve efficiency and effectiveness. The router for managing the layer skipping is expected to skip layers that have redundancy. The skipped layer is substitute by an adapter for mitigating feature gap.

**Strengths:**

1. The paper is well-written and easy to follow

2. The problem solved is interesting and meaningful

3. The proposed method seems to be interesting and effective

**Weaknesses:**

1. Even though the paper is motivated through MoE, the method is more focusing on layer skipping, which is a generally well-studied field for LLM. There should be a subsection in the related work talking about this field. Moreover, these two papers [1,2] seem to be very relevant and should be compared or discussed. The current version makes it hard to judge the novelty or contribution.

[1] Raposo, David, et al. "Mixture-of-Depths: Dynamically allocating compute in transformer-based language models." arXiv preprint arXiv:2404.02258 (2024).
[2] Tan, Zhen, et al. "DLO: Dynamic Layer Operation for Efficient Vertical Scaling of LLMs." arXiv preprint arXiv:2407.11030 (2024).

2. The major motivation lie in the feature redundancy of layers in the MLLM, as shown in Fig 1. Can the author plot similar figures for the learned RoE model, to show that the redundancy is mitigated?

3. There seem to be no direct supervision signal for calculating the feature similarity and guiding the learning of the router. How to make sure the skipped layers are indeed redundant? Also, can the paper show the training loss to indicate that the convergence of the method?

4. Even though the paper focuses on VLLMs, the major design seems can also be applied to LLMs. It would be interesting to see how this will impact LLMs.

5. Since the redundancy is highly correlated to the hardship of the input instance, how to decide the sparsity before the training? If it's a hyper-parameter for tuning across datasets / tasks, then this might heavily impact the applicability of the proposed method for unseen tasks. Can the authors provide some insights on how to choose the sparsity? Also, the current tasks are more focusing on easier tasks like VQA. Is the method still effective / necessary for newer or harder tasks and benchmarks like grounding or segmentation?

6. Since the efficiency is the major target, can the author provide comparison of actual averaged FLOPs in the experiments, to explicitly show the effectiveness and importance of the proposed method?

**Questions:**

Please see above.

---

> ### Author Response · Authors · 2024-11-23
>
> # Comment to Reviewer PWB7
>
> We highly appreciate your time and effort in reviewing this paper, as well as your encouraging and constructive comments on our work. Below, we response to your key concerns point by point.
>
> **Comment 1:** Even though the paper is motivated through MoE, the method is more focusing on layer skipping, which is a generally well-studied field for LLM. There should be a subsection in the related work talking about this field. Moreover, these two papers [1,2] seem to be very relevant and should be compared or discussed. The current version makes it hard to judge the novelty or contribution.
>
> [1] Raposo, David, et al. "Mixture-of-Depths: Dynamically allocating compute in transformer-based language models." arXiv preprint arXiv:2404.02258 (2024).
>
> [2] Tan, Zhen, et al. "DLO: Dynamic Layer Operation for Efficient Vertical Scaling of LLMs." arXiv preprint arXiv:2407.11030 (2024).
>
> **Response:** Thank you for your suggestion and recommending these excellent works [1,2].
> Except the principle of dynamic inference, our RoE mainly differs from these works in the following aspects:
>
> MoD [1] focuses on token-wise redundancy, while our RoE is a layer-wise one.
> In particular, MoD evaluates the importance of tokens and drops the less important ones during inference.
> We think that the two methods have orthogonal contributions and are likely to be compatible with each other.
>
> In terms of DLO [2], its target is different from our RoE, which aims to achieve effective vertical expansion of LLM. In practice, DLO involves the expansion of Transformer layers, and its layer activation refers to the MLP modules. We can see that efficiency is not the top priority in DLO, which often yields more computation than the default model.
>
> Compared with these two excellent works, we mainly focus on the layer redundancy issues of exiting MLLMs, and also tackle with several key challenges of dynamic layer skipping, such as feature gap, sparsity optimization and the alignment of training and testing.
>
> **Comment 2:** The major motivation lie in the feature redundancy of layers in the MLLM, as shown in Fig 1. Can the author plot similar figures for the learned RoE model, to show that the redundancy is mitigated?
>
> **Response:** Thanks for this comment.
> Following your suggestion, we visualize more samples and calculate their L1-distance in each layer.
> The orange bar represents the skipped transformer layer.
> The visualizations are given in our anonymous project: https://anonymous.4open.science/r/RoE_Rebuttal-5A77/l1-norm.png.
> From the samples, we can see that RoE can help the model skip layers with low L1-distance scores, suggesting the less importance or more redundancy of these layers.
>
> **Comment 3:** There seem to be no direct supervision signal for calculating the feature similarity and guiding the learning of the router. How to make sure the skipped layers are indeed redundant? Also, can the paper show the training loss to indicate that the convergence of the method?
>
> **Response:** Thanks for this insightful question.
> Similar to previous dynamic models like MoE [3,4], we use the default prediction loss to optimize the router.
> In principle, its optimization target is to minimize the negative impact of skipped layerst on performance, so that the router are teached to judge the importance/redundancy of the layers for different examples.
>
> During training, we will combine the outputs of both the adapter and the Transformer layer based on the predicted weights of router, *i.e.*, Eq.10.
> In this case, the gradients can be back-propagated to optimize the router.
>
> As discussed in the paper, this indirect optimization is not easy to achieve the desired sparsity target, since the MLLMs are already well trained.
> Thus, we introduce the sparsity regularization to enforce the learning of routers.
> In particular, when the skipping ratio is less than the target, the sparsity regularization will enforce the routers to increase the weight of skipping, as described in Eq.7.
>
> The training losses of RoE are plotted and given in our anonymous project: https://anonymous.4open.science/r/RoE_Rebuttal-5A77/training_loss.png.
> At the router warmup stage, *i.e.* 750 to 1250 step, we can observe that the training loss first drops rapidly, and then enters a state of convergence oscillation at the 900-th step, suggesting the routers are well optimized.
>
> [3] Lin, Bin, et al. "Moe-llava: Mixture of experts for large vision-language models." arXiv preprint arXiv:2401.15947 (2024)
>
> [4] Chen, Shaoxiang, et al. "Llava-mole: Sparse mixture of lora experts for mitigating data conflicts in instruction finetuning mllms." arXiv preprint arXiv:2401.16160 (2024)

---

> > ### Author Response · Authors · 2024-11-23
> >
> > **Comment 4:** Even though the paper focuses on VLLMs, the major design seems can also be applied to LLMs. It would be interesting to see how this will impact LLMs.
> >
> > **Response:** Thanks for this comment.
> > Following your suggestion, we apply our RoE to LLaMA-3 8B on Alpaca-EN [5], and report the results in the following table.
> >
> > |Method|Skip Ratio|Commonsense QA|Skip Ratio|Logiqa|Skip Ratio|WNLI|
> > |-|-|-|-|-|-|-|
> > |LLaMA-3 8B|0.0%|76.25|0.0%|27.65|0.0%|61.97|
> > |RoE_30%|15.8%|75.10|6.42%|29.03|3.83%|60.56|
> >
> > It can be seen that our RoE is also applicable to representative LLMs.
> > For instance, retaining about a 15.8% skipping rate, the performance on Commonsense QA is barely reduced, *i.e.*, 1.15%.
> > Notably, this trial is not carefully tuned due to the time limit, which still has ample room to improve.
> >
> > [5] Rohan Taori, et al. "Stanford Alpaca: An Instruction-following LLaMA model" GitHub repository (2023)
> >
> > **Comment 5:** Since the redundancy is highly correlated to the hardship of the input instance, how to decide the sparsity before the training? If it's a hyper-parameter for tuning across datasets / tasks, then this might heavily impact the applicability of the proposed method for unseen tasks. Can the authors provide some insights on how to choose the sparsity? Also, the current tasks are more focusing on easier tasks like VQA. Is the method still effective / necessary for newer or harder tasks and benchmarks like grounding or segmentation?
> >
> > **Response:** Thanks for your careful review.
> > Similar with MoE [3] and MoD [1], the sparsity target of RoE is also empirically set during training, but its mechanism is closer to your mentioned dynamic inference, *i.e.*, the redundancy is related to the hardship of input instance.
> >
> > To explain, our sparsity regularization is a soft constraint for MLLMs, while the ones of MoE and MoD are hard settings.
> > Concretely, MoE needs to carefully set its sparsity configurations before training, *e.g.*, the number and sizes of its experts, and its inference cost is the same for different examples regardless of the example hardship.
> > In terms of MoD, the fixed pruning ratios are set for all layers of an LLM, which are strictly executed for all examples during training.
> >
> > In contrast, our RoE poses a soft target for MLLM training, *i.e.*, encouraging the model to achieve a desired sparsity rate in a training batch, rather than implementing hard constraints on each example.
> > Thus, we can see that during testing, RoE can select the routing paths of varying lengths for different examples, as shown in Fig.3, which is also consistent with your comment that redundancy is highly correlated to the hardship of the input instance.
> > Moreover, the tuning expenditure of RoE is relatively cheap for MLLMs requiring no VL pretraining, so its empirical setts are actually affordable.
> >
> > In terms of generalization to unseen tasks, we think that it is an intrinsic problem of MLLMs.
> > Since the training data scale is much smaller than LLM, most MLLMs often show limited generalization ability to unseen tasks, especially your mentioned grounding or segmentation tasks with distinct output formats from common VL tasks.
> > For instance, without tuning on grounding data, LLaVA cannot output the coordinates of the referred objects in the instructions.
> >
> > **Comment 6:** Since the efficiency is the major target, can the author provide comparison of actual averaged FLOPs in the experiments, to explicitly show the effectiveness and importance of the proposed method?
> >
> > **Response:** Thanks for your suggestion.
> > We report the reduction of TFLOPs in the following table.
> >
> > |Method|SQA|GQA|MMB|POPE|
> > |-|-|-|-|-|
> > |LLaVA|9.8|8.8|9.5|8.8|
> > |RoE-LLaVA_{10\%}|8.4 (-13.9%) |8.0 (-8.9%)  |7.2 (-23.9%) |7.6 (-13.2%) |
> > |RoE-LLaVA_{20\%}|7.5 (-23.8%) |7.7 (-12.0%) |6.9 (-26.9%) |6.3 (-28.0%) |
> > |RoE-LLaVA_{30\%}|7.2 (-26.2%) |7.7 (-13.0%) |6.5 (-31.9%) |5.8 (-33.6%) |
> > |VILA|9.8|8.8|9.5|8.8|
> > |RoE-VILA_{10\%}|8.5 (-12.9%) |8.0 (-9.2%)  |8.1 (-14.4%) |7.4 (-15.4%) |
> > |RoE-VILA_{20\%}|7.2 (-27.0%) |7.4 (-16.0%) |7.3 (-23.0%) |6.5 (-25.9%) |
> > |RoE-VILA_{30\%}|7.0 (-28.2%) |7.3 (-17.3%) |6.6 (-30.6%) |6.1 (-30.9%) |
> >
> > It can be seen that RoE can significantly reduce computation complexity.
> > For instance, RoE-LLaVA_{30%} can reduce FLOPs by up to 31.9% on MMB benchmarks.

---

> > > ### Comment · Reviewer_PWB7 · 2024-11-26
> > > **Response to rebuttal**
> > >
> > > Thanks for the rebuttal. It partially solved my questions. However, I still have concerns on 1. the overlook of a related work section on model / llm layer skipping literature, such as [1,2], making the contribution unclear. 2. The lack of experiments on newer VLLMs that are capable for more complex tasks such as grounding, segmentation, etc. I would raise my score a bit.
> > >
> > > [1] Wang, Jue, et al. "Skipbert: Efficient inference with shallow layer skipping." Proceedings of the 60th Annual Meeting of the Association for Computational Linguistics (Volume 1: Long Papers). 2022.
> > > [2] Hu, Boren, et al. "SmartBERT: a promotion of dynamic early exiting mechanism for accelerating BERT inference." Proceedings of the Thirty-Second International Joint Conference on Artificial Intelligence. 2023.

---

> > > > ### Author Response · Authors · 2024-11-27
> > > >
> > > > Thanks a lot for your reply. We respond to your new concerns below.
> > > >
> > > > **Comment 7:** The overlook of a related work section on model / llm layer skipping literature, such as [1,2], making the contribution unclear.
> > > >
> > > > [1] Wang, Jue, et al. "Skipbert: Efficient inference with shallow layer skipping." Proceedings of the 60th Annual
> > > > Meeting of the Association for Computational Linguistics (Volume 1: Long Papers). 2022.
> > > >
> > > > [2] Hu, Boren, et al. "SmartBERT: a promotion of dynamic early exiting mechanism for accelerating BERT
> > > > inference." Proceedings of the Thirty-Second International Joint Conference on Artificial Intelligence. 2023.
> > > >
> > > > **Response:** Thank you for your suggestion and recommending these excellent works [1,2].
> > > > Compared with previous efforts, we mainly differs in the following aspects:
> > > >
> > > > 1)  SkipBERT [1] focuses more on the shallow layers of BERT,
> > > > while RoE tends to skip the higher layers of MLLMs. To explain,
> > > > in pretrained language models like BERT,
> > > > their shallow layers often attend to word tokens nearby,
> > > > so that SkipBERT can use *n-gram* to replace them for better efficiency.
> > > > In contrast, RoE reveal that the shallow layers of MLLMs are critical for multimodal fusion,
> > > > and more redundancy exits in the higher layers, as shown in Fig.1.
> > > >
> > > > 2) SmartBERT [2] is a classical dynamic network combing layer-skipping with early-exit.
> > > > In addition to the difference in routing features, *i.e.*, [cls] token for SmartBERT and new routing token for RoE,
> > > > the training strategies of two methods are also greatly different.
> > > > Besides, our RoE also introduce novel designs like adapter-based skip connection and structural regularization for feature gap and model sparsity, respectively.
> > > >
> > > > More importantly, RoE realizes the dynamic inference of exiting MLLMs, which differs from BERT-like language models in both parameter scale and task difficulty.
> > > > Thus, we believe that our contributions and novelties are orthogonal but notable.
> > > >
> > > > Following your suggestion, we will add the discussions with previous layer skipping literature in our final submission.
> > > >
> > > > **Comment 8:** The lack of experiments on newer VLLMs that are capable for more complex tasks such as grounding, segmentation, etc. I would raise my score a bit.
> > > >
> > > > **Response:** Thanks for this suggestion.
> > > > For a quick response to your concern, we directly examine RoE-LLaVA and RoE-VILA on the visual grounding task, and report the results in the following table.
> > > >
> > > > |Method|Skip Ratio|RefCOCO textA|Skip Ratio|RefCOCO textB|Skip Ratio|refcoco+ testA|Skip Ratio|refcoco+ testB|
> > > > |-|-|-|-|-|-|-|-|-|
> > > > |LLaVA|0.0%|64.4|0.0%|47.5|0.0%|59.2|0.0%|39.0|
> > > > |RoE-LLaVA|0.42%|65.3|0.26%|48.9|0.53%|60.8|0.29%|40.0|
> > > >
> > > > |Method|Skip Ratio|RefCOCO textA|Skip Ratio|RefCOCO textB|Skip Ratio|refcoco+ testA|Skip Ratio|refcoco+ testB|
> > > > |-|-|-|-|-|-|-|-|-|
> > > > |VILA|0.0%|69.8|0.0%|52.1|0.0%|64.0|0.0%|43.9|
> > > > |RoE-VILA|0.13%|70.7|0.13%|52.8|0.16%|65.2|0.15%|44.9|
> > > >
> > > > It can be seen that our RoE is also applicable to the visual grounding task without specific tuning.
> > > > But we also note that the actual skip rate is not very significant.
> > > > We attribute this case to two main factors.
> > > > First, most MLLMs like LLaVA and VILA are mainly trained for knowledge-based QA, and we can see that grounding task has different principles and output formats from common VL tasks.
> > > > Second, the grounding examples only makes up a small portion of MLLMs, e.g., about 16%.
> > > > Also due to the joint seq2seq prediction,
> > > > their actual performance are even much worse than previous bespoke small models [3,4].
> > > >
> > > > In this case, we can conclude that visual grounding is a challenging task for VILA and LLaVA.
> > > > According to our arguments, RoE will skip fewer layers for the ``difficulty'' examples, also as discussed above.
> > > > However, the principle and generalization of our RoE can be still confirmed.
> > > >
> > > > Thanks again for your valuable time and efforts, and we look forward to your further discussions.
> > > >
> > > > [3] Gen Luo, et al. "Multi-task Collaborative Network for Joint Referring Expression Comprehension and Segmentation." IEEE/CVF Conference on Computer Vision and Pattern Recognition (2020)
> > > >
> > > > [4] Luo, Gen, et al. "A Survivor in the Era of Large-Scale Pretraining: An Empirical Study of One-Stage Referring Expression Comprehension." IEEE Transactions on Multimedia (2023).

---

> ### Author Response · Authors · 2024-11-25
>
> Dear Reviewer PWB7,
>
> Thanks again for your great efforts and constructive advice in reviewing this paper! With the discussion period drawing to a close, we expect your feedback and thoughts on our reply. We put a significant effort into our response, with several new experiments and discussions. We sincerely hope you can consider our reply in your assessment.
>
> We look forward to hearing from you, and we can further address unclear explanations and remaining concerns if any.
>
> Regards,
>
> Authors

---

### Meta-Review · Area_Chair_SLWy · 2024-12-22

**Metareview:**

### Claims and Findings:
  - The paper proposes a method to enhance the efficiency of multimodal large language models (MLLMs) by selectively skipping certain transformer layers. These skipped layers are linked to a low-rank adaptor. A learned routing function determines whether a layer should be skipped based on the input. This approach can accelerate inference time while retaining similar performance.

### Strengths:
   - The proposed method is intuitive and well-founded.
   - The experimental results convincingly demonstrate the method's performance, showing consistent speedups in MLLMs and surpassing existing MOE methods in accuracy on downstream tasks.

### Weaknesses:
   - A more comprehensive discussion of related works is needed to better position the proposed work and help reviewers understand its novelty.
  - Additional ablation studies, experiments on complex tasks such as grounding and segmentation, and a discussion of the accuracy/speed trade-off are necessary to fully demonstrate the benefits of the method.
  -  More attention to details and improvements in the paper's clarity are needed.

### Reasons for Decision:
  - Based on the identified strengths of the paper.

**Additional Comments On Reviewer Discussion:**

The authors have effectively addressed the reviewers' concerns in their rebuttal by providing detailed explanations, clarifying the novelty of the work and some technical details, conducting additional experiments and ablation studies, and discussing related works and the accuracy/speed trade-off.

Following the rebuttal, three reviewers raised their ratings from 5 to 6, and all four reviewers are inclined to accept the paper.

The AC recommends acceptance of the paper but notes that the authors have not yet incorporated the rebuttal discussion into the paper, which should be done in the final version.

---

### Decision · Program_Chairs · 2025-01-22

Accept (Poster)